# Low-Platinum-Content Exchange-Coupled CoPt Nanoalloys with Enhanced Magnetic Properties

**DOI:** 10.3390/nano14060482

**Published:** 2024-03-07

**Authors:** Georgia Basina, Vasileios Alexandrakis, Ioannis Panagiotopoulos, Dimitrios Niarchos, Eamonn Devlin, Margarit Gjoka, George C. Hadjipanayis, Vasileios Tzitzios

**Affiliations:** 1Institute of Nanoscience and Nanotechnology, National Centre for Scientific Research “Demokritos”, 15310 Athens, Greece; v.alexandrakis@inn.demokritos.gr (V.A.); d.niarchos@inn.demokritos.gr (D.N.); m.gjokas@inn.demokritos.gr (M.G.); 2Department of Physics and Astronomy, University of Delaware, Newark, DE 19716, USA; hadji@udel.edu; 3Department of Materials Science and Engineering, University of Ioannina, 45110 Ioannina, Greece; ipanagio@uoi.gr; 4Amen New Technologies, 15343 Athens, Greece; 5Department of Chemical Engineering, Northeastern University, Boston, MA 02115, USA

**Keywords:** cobalt–platinum alloy, L1_0_ phase, CoPt, Co_3_Pt, exchange coupling, bimetallic nanoparticles, polyol method

## Abstract

Bimetallic colloidal CoPt nanoalloys with low platinum content were successfully synthesized following a modified polyol approach. Powder X-ray diffraction (XRD), Fourier transform infrared spectroscopy (FT-IR), thermogravimetric analysis (TGA), and transmission electron microscopy (TEM) studies were performed to estimate the crystal structure, morphology, and surface functionalization of the colloids, respectively, while the room-temperature magnetic properties were measured using a vibrating sample magnetometer (VSM). The particles exhibit excellent uniformity, with a narrow size distribution, and display strong room-temperature hysteretic ferromagnetic behavior even in the as-made form. Upon annealing at elevated temperatures, progressive formation and co-existence of exchange coupled, of both chemically ordered and disordered phases significantly enhanced the room-temperature coercivity.

## 1. Introduction

Magnetic nanoparticles represent one of the most studied classes of nanoparticles because of their size-dependent magnetism, especially in the nanoscale regime, and their potential applications in many fields [1,2,3,4,5,6]. Consequently, a lot of research effort focuses on the development of synthetic methodologies to precisely control their size, shape, composition, crystal structure, and surface chemical modification [6,7,8,9].

Chemically ordered, bimetallic CoPt nanoparticles are extensively studied due to their technological interest in a variety of fields such as permanent magnets, high-density magnetic storage media, catalysis, and biomedicine [10]. The CoPt phase diagram includes several chemically ordered phases as Co_3_Pt, CoPt, and CoPt_3_ [11,12]. The equiatomic CoPt, when chemically ordered, forms the L1_0_ high magnetic anisotropy (K_u_ = 4.9 MJ/m^3^) tetragonal phase with alternating atomic layers of Co and Pt along the c axis. This phase is reported to exist in the range of 40–60 at.% platinum composition. The off-equiatomic stoichiometry compositions can be only partially ordered and consequently have somewhat lower anisotropies. On the Pt-rich and the Co-rich sides, the CoPt_3_ and Co_3_Pt are formed, respectively, which are both chemically ordered cubic L1_2_ phases and have moderate magnetic anisotropy [13,14].

The high anisotropy values of these CoPt phases [15,16] allow for the stabilization of their magnetization against thermal fluctuations and demagnetizing effects at very low dimensions [17,18,19,20,21], properties which are fundamental for application in permanent magnets and recording media [22,23,24,25]. Additionally, their chemical stability in alkaline and acidic environments makes them excellent candidates for low-platinum-content electrocatalysts [26,27,28,29,30,31,32,33,34].

One of the major commercialization issues that still remains is the high precious metal content, up to 50 at.%, equivalent to more than 77 wt.%, which is necessary, as previously mentioned, to form the chemically ordered L1_0_ structure. Therefore, there is still a great challenge in the synthesis of CoPt bimetallic alloys, with Pt content that is as low as possible, maintaining high magnetocrystalline anisotropy, and consequently high coercivity. Previous studies have presented a plethora of chemical routes for the synthesis of ultrafine, monodispersed, bimetallic CoPt nanoparticles [35,36,37,38,39,40,41]. The as-made nanoparticles, in a majority of the studies, pose the disordered face-centered cubic (*fcc*) crystal structure and show superparamagnetic behavior. Our group recently published the effect of bismuth addition on the equiatomic CoPt L1_0_ ordering, in which the particles show partial ordering without any post-annealing exhibiting a 1.7 kOe room-temperature coercivity [42]. In general, thermal treatment at elevated temperatures is required to obtain the fully ordered, magnetically hard L1_0_ crystalline phase. The bulk order–disorder transition of CoPt occurs at 825 °C [43], while in the nanosize regime, ordering can take place at temperatures above 650 °C [44,45]. Regarding the synthesis of the L1_2_ Co_3_Pt phase, there is a limited number of works in the literature, and the majority of them are related to synthesis in thin film form [9,46,47,48,49,50,51].

Here, we report the colloidal synthesis of platinum lean CoPt bimetallic nanoalloys, following a modified polyol methodology. The materials reveal high uniformity, narrow size distribution, and exhibit enhanced room-temperature coercivity, up to 14.5 kOe, a value which is among the highest in the literature concerning CoPt-based nanoalloys of any composition. The enhanced magnetic properties result from the coupling between the magnetically hard L1_0_ and semi-hard Co_3_Pt phases, leading to the formation of an exchange-spring nanocomposite magnetic material.

## 2. Materials and Methods

### 2.1. Materials Synthesis

Anhydrous Co(CH_3_COO)_2_, (Cobalt(II) acetate, anhydrous, 98+%, Thermo Scientific Chemicals, Waltham, MA USA), and PtCl_4_, (Platinum(IV) chloride, Sigma-Aldrich, St. Louis, MO, USA), were used as cobalt and platinum precursors, while polyethylene glycol 200 (PEG-200, Sigma-Aldrich) served as both a solvent and a reducing agent. Equimolecular quantities of oleic acid and oleyl amine were utilized as nanoparticles’ capping agents. The synthesis was carried out following the chemical approach used in our previous work for the synthesis of equiatomic L1_0_ CoPt nanoalloys [44]. Briefly, a 100 mL spherical flask was charged with a mixture of 20 mL PEG-200, 2 mmol oleic acid, (technical grade, 90%, Sigma-Aldrich), and 2 mmol oleyl amine, (technical grade, 70%, Sigma-Aldrich), and the temperature was raised to 100 °C and degassed under nitrogen bubbling for 1 h, followed by the addition of PtCl_4_ (0.5 mmol) and anhydrous Co(CH_3_COO)_2_ (1.66 mmol) under vigorous magnetic stirring. The nanoalloys synthesis reaction was performed at 250–260 °C under N_2_ mantle for 1 h. The colloidal particles were precipitated by the addition of ethanol and separated by centrifugation. The excess of oleic acid, oleyl amine, PEG, and reaction byproducts, were rinsed by repeated washing with ethanol and separation by centrifugation. Finally, the particles were dried at room temperature under reduced pressure and the reaction yield was estimated gravimetrically. The equiatomic CoPt nanoalloys were synthesized using 0.5 mmol of each metallic precursor following exactly the above methodology.

### 2.2. Materials Characterization

Powder X-ray diffraction (XRD, Siemens D 500 diffractometer with Cu Kα radiation, Siemens, Berlin, Germany) was performed for the crystal structure analysis, and the morphology of the nanoparticles was estimated using transmission electron microscopy (TEM, JEOL JEM-3010, JEOL Ltd., Tokyo, Japan). The surface functionalization was estimated by Fourier transform infrared spectroscopy (Bruker FT-IR spectrometer, Equinox 55/S model, Bruker, Billerica, MA, USA) while thermogravimetric analysis (Perkin-Elmer Pyris Diamond TGA/DTA, PerkinElmer, Waltham, MA, USA) was conducted to quantify the organic content. Magnetic measurements were measured at room temperature using a vibrating sample magnetometer (VSM) equipped with a 2 T magnet.

## 3. Results and Discussion

The size and the morphology of the platinum lean CoPt nanoalloys in both as-made and extensively annealed (700 °C, 7 h) forms were studied by TEM microscopy (Figure 1). The as-made nanoparticles were uniform and quite monodispersed, while after annealing at 700 °C for 7 h under H_2_/Ar flow, the particles became slightly irregular without significant sintering and agglomeration. The as-made particles were well spherically shaped with 7.1 nm mean diameter and narrow size distribution, while the annealed material lost its spherical morphology and had an average size of 15.7 nm (size distribution histogram is presented in the Appendix A, Appendix A).

Figure 2a illustrates the infrared (IR) spectrum of CoPt nanoparticles capped with oleic acid–oleyl amine. The spectrum is collected in CCl_4_ colloidal solution in a liquid cell and shows strong bands at 2854 and 2928 cm^−1^, assignable to the symmetric and asymmetric CH_2_ stretches of the hydrocarbon moiety, respectively; a shoulder at 2960 cm^−1^, due to the asymmetric stretch of the terminal CH_3_ group; and a weak yet definite band at 3006 cm^−1^, which is attributed to the olefinic CH stretch from both oleic acid and oleyl amine molecules [44,52,53]. The broad bands around 3400 cm^−1^ are attributed to the NH_2_ group, confirming the presence of oleyl amine on the nanoparticle surface [44]. Additionally, two other absorptions at 1569 cm^−1^ and 1412 cm^−1^ are characteristic of the presence of carboxylate (-COO-) groups. Moreover, the frequency separation between the bands at 1569 cm^−1^ and 1412 cm^−1^ indicates that the oleate moieties bind the surface atoms in a chelating or bridging mode of coordination [54]. Finally, the band at 1464 cm^−1^ is associated with the CH_2_ deformation (ν*_CH_*_2_). These data clearly demonstrate the anchoring of both capping agents to the nanoparticle’s surface. The as-made oleic acid–oleyl amine-capped CoPt nanoparticles are easily dispersible in non-polar organic solvents such as hexane, toluene, and chloroform, with concentrations of up to several dozen mg/mL, and are stable for weeks without precipitation. Thermogravimetric analysis was used to determine the mass of the organic matter absorbed on the CoPt nanoparticles (Figure 2b). The experiments were carried out under a nitrogen atmosphere with 60 mL/min flow and 5 °C/min heating rate. The organic molecules’ desorption begins around 140–150 °C and the total weight loss is 28%. At temperatures of about 360–450 °C, the curve approaches an intermediate plateau. This behavior suggests, in agreement with the literature [55], that the weight loss from 140 to 360 °C is due to the desorption of the amine ligand, and the weight loss above 360 °C is mainly due to the desorption/decomposition of carboxylic residue. The overall desorption procedure was completed at approximately 500 °C. The reaction yield was estimated gravimetrically. The dry CoPt powder from a single batch was weighed as 254 mg. Considering that 28 wt. % belongs to the organic capping molecules, the net mass of CoPt nanoparticles is 182.9 mg, resulting in a reaction yield of up to 93.6%. Assuming that the Pt^2+^ reduces quantitatively, due to the positive reduction potential (+1.18 V) in contrast with the Co^2+^ (−0.282 V), the estimated nanoalloys’ atomic composition is Co_74.4_Pt_25.6_, which is very close to the nominal composition (Co_77_Pt_23_), according to the reaction precursors molecular quantities.

Figure 3A shows the powder XRD patterns of the as-made as well as the annealed (at 700 °C) bimetallic CoPt nanoalloys. The pattern of the as-made material (Figure 3A(a)) shows clearly that the dominant phase is the chemically disordered *fcc*. As for the reflection at 2-theta = 31.7 degrees, which remains unmatched and is indicated with an asterisk, it can be attributed to the cobalt oxide phase, probably the Co_3_O_4_ [56]. Furthermore, the broad peaks correspond to a Scherrer structural coherence size of 6 nm in diameter, which is slightly smaller than the particle size that was estimated by the TEM studies (7.1 nm). It is worth mentioning that the large unit cell, a = 3.88 Å, shows a Pt-rich phase [8,57]. This behavior has been previously reported by our group [51] for similar materials and due to a strong compositional gradient from the nanoparticles’ core to the shell, indicating the presence of a Pt-rich composition at the center of the particle, which progressively decreases, and leads to a Co-rich shell and can be explained by the reaction conditions. PEGs, without an alkaline environment, as well as oleyl amine are mild reducing agents that are not able to reduce Co^2+^ to the metallic state [58]. The presence of platinum ions in the reaction—which, due to their electropositivity is reduced very easily without the need for strong reducing agents—leads to the initial formation of tiny Pt seeds, possessing very negative redox potential [59], which can consequently reduce much more easily the Co^2+^ to the metallic state, leading to the formation of particles with Pt-rich cores [57]. We have also noticed that the chemical ordering requires a short-length scale diffusion process, which means that this compositional gradient can be present even if chemical ordering to various phases PtCo-Pt_0.4_Co_0.6_-PtCo_3_ (from center to shell) has been reached. After thermal treatment at 700 °C for 4 h, the X-ray diffraction pattern in Figure 3A(b) reveals mixtures of structures consisting of the following phases: A tetragonal phase with a = 3.79 Å and c = 3.68 Å indicated by the presence of (001)~24°, (110)~33.2°, superlattices, and additionally, the presence of (111)~41.7°, (200)~47.9° and (002) at 49.3°. The lattice constants of this tetragonal phase are those expected for the stoichiometric fully ordered L1_0_ CoPt alloy [56,60,61]. On the other hand, the weaker diffraction at 40.5° can be assigned to the (111) of cubic structured CoPt binary alloy. In the case of annealing for a longer time, i.e., 700 °C for 7 h, the shifting and splitting of the main diffractions of the as-made nanoparticles shows that part of the alloys transformed to the ordered L1_0_ and cubic L1_2_ Co_3_Pt phases [62,63]. Rietveld analysis (Appendix A) shows that the tetragonal phases possess 11.4%, while the 88.6% belong to the presence of cubic Co_3_Pt phases. Taking into account the presence of off-stoichiometry phases as well as the low percentage of the tetragonal phase, we can interpret the fact that the overall composition could be close to a 3/1 cobalt-to-platinum atomic ratio.

Detailed examination of the powder X-ray diffraction pattern in Figure 3A(c) shows the obvious existence of the (001) and (110) superlattice peaks, which indicate the transformation to the chemically ordered tetragonal L1_0_ phase [57,60,61], while some extra peaks indicate the presence of another phase with weak tetragonality: *a* = 3.73 Å and *c* = 3.71 Å indicated by the presence of (111)~41.88° and (200)~48.8°, (002)~49.7°. Since the order parameter *S* (Equation (1)) scales with the tetragonicity as *S*^2^~ (1 − c/a), the c/a = 0.995 compared with the 0.973 of the fully ordered phase gives *S* = 0.37.
(1)$$S^{2}=\frac{1-c/a}{1-{\left(c/a\right)}_{f}}$$
where *(c*/*a)_f_* is the ratio for the fully ordered phase and *c*/*a* for the partially ordered phase. The use of this expression is justified by the fact that the order parameter can be proved to scale $S^{2}=\left(1-c/a\right)\left(B/4\right)$, where the parameter *B* can be considered independent of *S*. This could be due to the off-stoichiometric composition of this phase as the lattice parameters are close to what is expected for Pt_0.4_Co_0.6_. The S of this cobalt-rich phase shows that the material could reach an anisotropy of 1.8 MJ/m^3^. Finally, the stronger peaks can be assigned to the cubic L1_2_ Co_3_Pt phases. For this phase, the Rietveld analysis (Appendix A) shows the presence of two different phases with *a* = 3.65 Å and *a* = 3.72 Å, respectively. The d-space value from the HR-TEM image (Figure 1f, 0.211 nm) can be assigned to the (111) plane of the Co_3_Pt phase, which is also very close to the results reported in the literature [16]. It should be noted that several peaks of the L1_2_ phase are located very close to those of L1_0_, making their detection difficult since they mask each other. Additionally, slight differences may also attribute to the existence of structural strains, which play a very important role in the case of nanostructured materials [64,65].

According to the phase diagram of bulk CoPt, cobalt-rich L1_2_ phase reaches its maximum ordering temperature at around 900 °C for the optimum stoichiometry, and decreases to 700 °C for Co_83_Pt_17_ [50,60,66,67]. Consequently, increasing annealing time at 700 °C is expected to increase the proportion of Co-rich L1_2_ phase, which is actually the dominant structure (88.4%). The lattice parameter, space group, and crystal structure of the CoPt sample annealed at 700 °C/7h are summarized in Appendix A, in the Appendix A. Regarding the crystallite size, it is evident that the reflection peaks become sharper and more intensive as the annealing temperature increases. This indicates that the particle size increases with longer annealing times. In particular, the size of the particles derived by the Scherrer equation after annealing at 700 °C for 4 h and 7 h was estimated to 8.8 nm and 17.1 nm, respectively. However, due to the morphological irregularity of the annealed materials, it is not possible to draw safe conclusions.

Room-temperature magnetic hysteresis loops of CoPt nanoalloys in the as-made and annealed forms have been performed in order to study their magnetic properties, and are presented in Figure 4. The as-made CoPt nanoparticles exhibit 59.3 emu/g saturation magnetization and a moderate coercivity (H_c_) value, around 1 kOe (Figure 4), which is significantly higher compared to the so-called “soft” magnetic materials where the coercivity is in the range of a few Oe. It is obvious that the as-made CoPt shows hysteretic ferromagnetic behavior which cannot be explained by the structural defects or imperfections (*pinning centers*). Shape anisotropy should also be excluded since the particles are almost perfectly spherical and nearly monodispersed as observed from the TEM images. Meanwhile, XRD measurements showed that the as-made nanoalloys consist predominantly of a disordered *fcc* phase, probably with the presence of a small amount of cobalt oxide. A possible interpretation should be connected with the presence of compositional gradient surface phenomena due to non-condensate atoms or exchange bias phenomena due to the presence of Co oxide in the as-made nanoalloys, which is also in agreement with the previous data discussed in the XRD section, as well as the reaction mechanism where the positive (+1.18 volts) Pt^2+^ reduction potential ensures the initial formation of tiny Pt seeds before the reduction in Co^2+^ ions. Under the regime of such a growth mechanism, it is expected that the nanoalloy core would be platinum-rich in contrast with the shell [44]. In the case of annealed nanoalloys, it is obvious from the hysteresis loops in Figure 4 that the magnetization value does not correspond to the saturation value, as the 2 T applied field is not sufficient to saturate the magnetization. Therefore, the magnetization at 2 T magnetic field is 46.8, 43.6, 36.8, and 33.7 emu/g for the samples after annealing at 700 °C for 30 min, 2, 4, and 7 h, respectively. The saturation values are also obtained by fitting the high-field data with the M = M_s_(1 − a/H) law (Appendix A). Extrapolation gives M_s_ = 36.4 emu/g, and therefore, M_R_/M_S_= 26.25/36.4 = 0.72.

On the other hand, the room-temperature coercivity reached 1.3, 4.2, 9.2, and 14.5 kOe after annealing at 700 °C for 30 min, 2, 4, and 7 h, respectively. Considering the unsaturated magnetization, it is expected that these values will be even higher. By examining the shape of the M vs. H curves, it is obvious that there are two different magnetic phases in the materials annealed for shorter times. This behavior becomes less prominent after 4 h annealing and almost disappears after 7 h annealing. For instance, in the sample annealed for 2 h, a significant change in the slope of M vs. H curve is observed, as indicated by the two local maxima of the derivative (Appendix A). In fact, the two maxima suggest two different magnetization switching mechanisms, which correspond to two different magnetic phases with two different coercive fields. This finding is in accordance with the conclusions from the XRD analysis, where different structural phases with different percentages of presence were identified according to Rietveld analysis. Indeed, the present system of nanoalloys is a mixture of cubic phases with intermediate magnetocrystalline anisotropy and ordered L1_0_ tetragonal CoPt phases exhibiting the highest magnetocrystalline anisotropy and coercivity [68]. As the annealing time increases, the degree of ordering and the proportion of the L1_0_ phase increase, which results in an enhancement of the coercivity value. It is evident from the hysteresis loop that, in addition to the increase in coercivity, the M vs. H curves become smoother, similar to single-phase magnetic hysteresis loops. This behavior is well known and based on the interactions between magnetically hard and soft phases and is described by the exchange-spring magnets theory [69,70]. It has been proven that the higher the difference in the magnetocrystalline anisotropy between the two phases, the lower the thickness (or the proportion) of the soft phase should be in order to have a completely coupled magnetic composite without steps at the demagnetization curve [71]. Therefore, the optimum conditions for effective coupling are more favorable under the presence of phases with intermediate hardness, as in graded stoichiometry nanoparticles. In nanostructured materials that need high-temperature annealing, it is very difficult to precisely control the dimensions of the hard phase and the soft phase in particular [70]. In our case, the presence of the Co_3_Pt phase, despite being a cubic phase, exhibits a moderate magnetocrystalline anisotropy (K_u_ = 10^7^ erg/cm^3^) [15]. It appears to be crucial for the formation of a fully exchange-coupled spring magnet with a single magnetization switching field (H_c_) as its progressive appearance with the annealing time completely eliminates the shoulder in the M vs. H curve. The importance of the Co_3_Pt phase formation, as well as the off-stoichiometric cobalt-rich L1_0_ phases, is also proven by the magnetic behavior of the nanoalloys when the annealing temperature is slightly decreased. After annealing at 675 °C, the magnetic hysteresis loops (Appendix A) show the presence of two different magnetic phases (a soft and a hard), which are much more intense compared to the sample annealed for the same time at 700 °C. The inefficient coupling may be attributed to the lower L1_0_ ordering, although we are convinced that it is also linked to the absence of the intermediate magnetocrystalline anisotropy Co-rich phase. Additionally, the remanence-to-saturation magnetization ratio (M_R_/M_S_) is enhanced well above the 0.5 value, which is expected for an isotropic sample. This indicates the presence of strong interactions. Moreover, the enhancement on the coercivity may also be attributed to the small particle size, especially in a multidomain region where magnetization changes by domain wall motion. The coexistence of both hard and semi-hard magnetic phases also arises from the uniform distribution of both elements in the materials, as shown in the HAADF STEM image of the CoPt sample annealed at 700 °C for 7 h, and the Co (red) and Pt (green) elemental mapping (Appendix A).

Furthermore, it is worth mentioning that, synthesizing equiatomic CoPt nanoalloys, i.e., following the same methodology but using equimolar cobalt and platinum precursors, shows that after annealing at 700 °C, we obtain a single-phase ferromagnetic material with a well-crystallized face-centered tetragonal phase (L1_0_) as presented in Appendix A in the Appendix A.

## 4. Conclusions

Bimetallic CoPt-based nanoalloys with low platinum content were successfully synthesized via a facile chemical methodology following a modified polyol process. The nanoparticles in the as-made form are monodispersed with a 7.1 nm mean diameter and unexpectedly reveal a high room-temperature coercive field, up to 1 kOe, which probably originated from the presence of gradients in the nanoalloys’ composition. After annealing at 700 °C, the nanoalloys progressively transformed into the fully and partial-ordered L1_0_ CoPt (hard) and cubic Co_3_Pt (semi-hard), respectively. The annealed nanoparticles exhibit an enhanced room-temperature coercive field, up to 14.5 kOe, which is amongst the highest values in the literature and with the lowest platinum content (25 at.%). The formation of the Co_3_Pt magnetically semi-hard phase is crucial in achieving a fully exchange-spring nanocomposite magnetic material. Therefore, the proposed platinum lean CoPt nanoalloys are very promising candidates for both permanent magnet and electrocatalytic applications.

## Figures and Tables

**Figure 1 nanomaterials-14-00482-f001:**
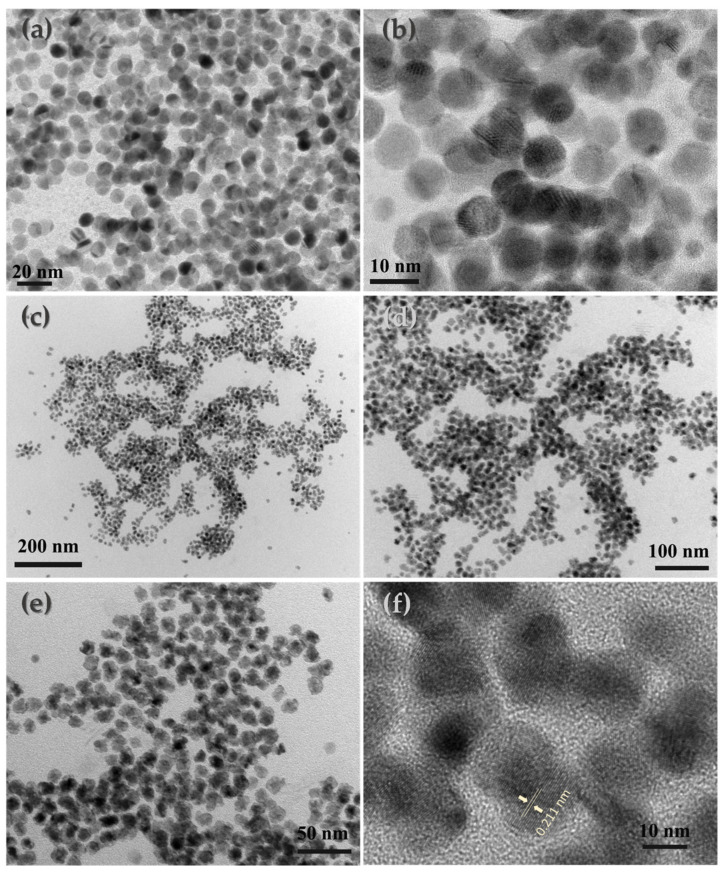
TEM and HR-TEM images of the as-made (**a**,**b**) and annealed at 700 °C for 7 h under H_2_-Ar atmosphere (**c**–**f**).

**Figure 2 nanomaterials-14-00482-f002:**
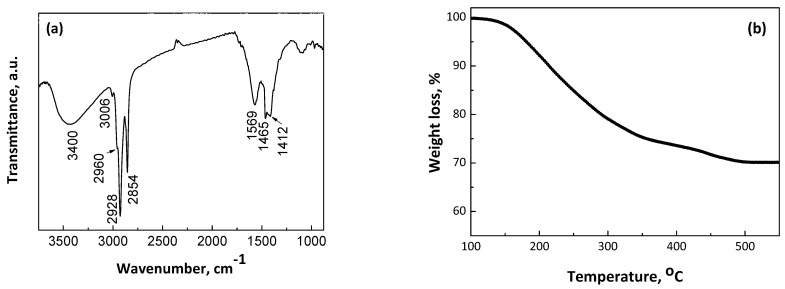
FT-IR spectrum (**a**) and TGA profile (**b**) from the as-made CoPt bimetallic nanoalloys.

**Figure 3 nanomaterials-14-00482-f003:**
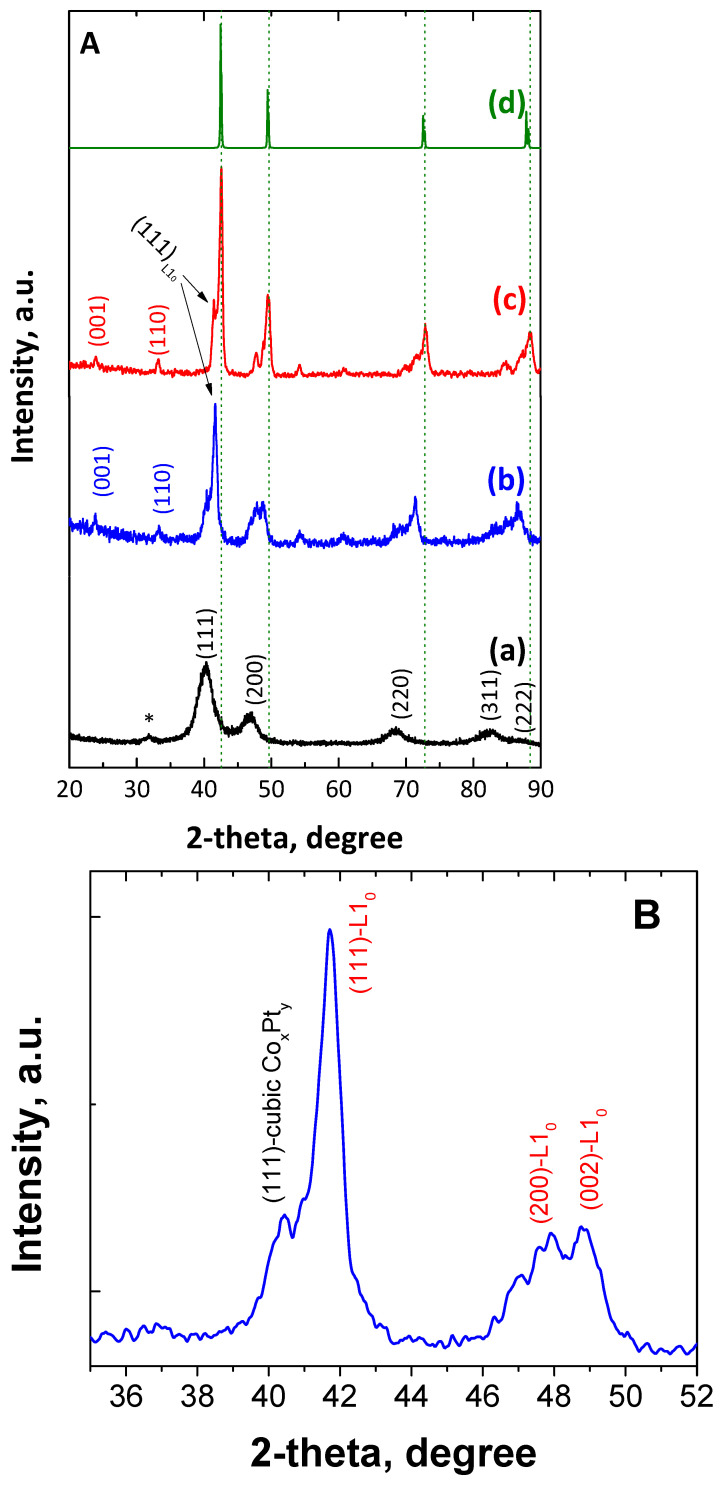
XRD patterns of CoPt nanoalloys (**A**) in the as-made form (**a**) and after annealing at 700 °C for 4 h (**b**), 7 h (**c**), and the calculated Co_3_Pt (**d**). Magnification of XRD patterns of CoPt nanoalloys (**B**,**C**) after annealing at 700 °C for 4 (blue line) and 7 (red line) h.

**Figure 4 nanomaterials-14-00482-f004:**
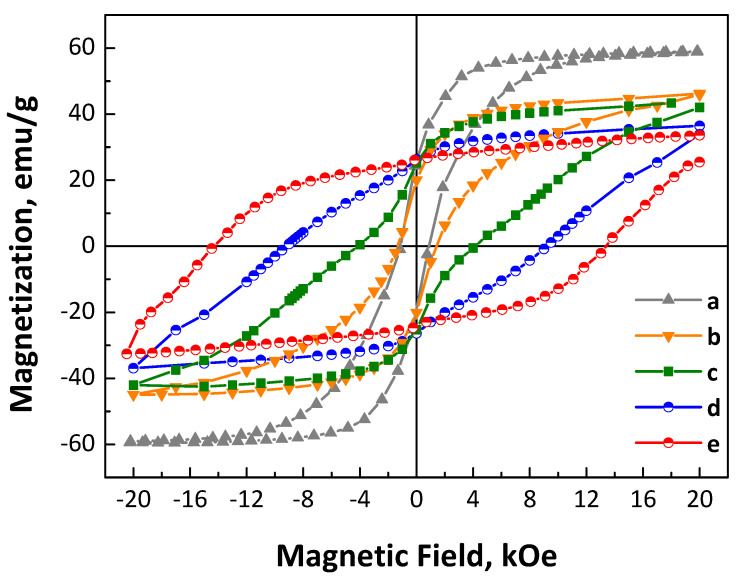
Room-temperature magnetic hysteresis loops of CoPt nanoalloys in the as-made (a) and after annealing at 700 °C for 30 min (b), 2 (c), 4 (d), and 7 h (e) under mild reductive atmosphere (4% H_2_ in Ar).

## Data Availability

Data are contained within the article.

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
