# Peer review of "Low-Platinum-Content Exchange-Coupled CoPt Nanoalloys with Enhanced Magnetic Properties"

_nanomaterials, 2024, doi:10.3390/nano14060482_

Round 1
Reviewer 1 Report (Previous Reviewer 3)
Comments and Suggestions for Authors
This is a very interesting work.
Author Response
Please see the attachment

Reviewer 2 Report (New Reviewer)
Comments and Suggestions for Authors
In this work, the authors study the CoPt nanoalloys with enhanced magnetic properties. Various characterizations are applied to analyze material’s crystal structure, morphology, surface functionalization, and ferromagnetic behavior. However, some important issues must be solved to improve this manuscript.
1. The descriptions should be rational, e.g., “a very low platinum content” for CoPt, which is actually not very low Pt content. Also, the abbreviations should be presented in the full names. Many mistakes about capital and small letters exist in the manuscript. Please check and revise these mistakes through the manuscript.
2. The authors should show the elemental distribution via EDX mapping to verify the uniform distribution of Co and Pt.
3. The necessary references for identifying the species in IR spectra should be provided.
4. The structural strain is an important parameter for material study, especially for the alloys. Obvious shift of XRD peaks is observed here, which is ascribed to the existence of structural strain. Please refer to j.jechem.2023.03.033 and 10.1063/5.0083059 for the analysis of structural strain.
5. Necessary electronic properties of Co and Pt should be measured and compared, e.g., XPS for Co and Pt.
Comments on the Quality of English LanguageModerate editing of English language required
Author Response
Please see the attachment

This manuscript is a resubmission of an earlier submission. The following is a list of the peer review reports and author responses from that submission.
Round 1
Reviewer 1 Report
Comments and Suggestions for Authors
Mahdaoui et al. synthesized bimetallic colloidal CoPt nanoalloys, with a very low platinum content following a modified polyol approach. Powder X-ray diffraction, IR spectroscopy, thermogravimetric analysis, and Transmission Electron Microscopy studies were performed to estimate the crystal structure, morphology and surface functionalization of the colloids respectively, while the room temperature magnetic properties were measured using Vibrating Sample Magnetometer. Overall, the idea received my attention and the methodology is technically sound. The topic is original and relevant in the field. It also addresses a specific gap in the field. However, there are some specific issues the authors should address by making modifications before we can proceed and positive action can be taken.
1. Compare the properties of your material to the state-of-the-art in literature and industry.
2. Please give a detailed methodology to facilitate reproducibility.
3. Please label each subfigure as a, b and c…
4. The results and discussion section can be divided into several subsections rather than presented in a huge section.
5. The authors mainly investigated the magnetic properties. Have the authors noticed some studies related to this point? i.e., [Magnetic behaviors of 3d transition metal-doped silicane: a first-principle study. J. Supercond. Nov. Magn. 2018, 31, 2789–2795, doi:10.1007/s10948-017-4532-4]…
Comments on the Quality of English LanguageThe English language requires improvements. Spelling and grammatical errors exist in the manuscript. i.e., l. 148: It is worth to mention that…; l. 161: much more easier; l. 211: previously discussion… We recommend you ask a native English speaker to edit the paper or use an independent professional editor.
Author Response
Reviewer 1:
Compare the properties of your material to the state-of-the-art in literature and industry.
We would like to thank the reviewer for the important observation. However, selected literature data on the room temperature magnetic properties of various CoPt-based nanoalloys, are provided for comparison in the Supplementary Material / Table S1.
Please give a detailed methodology to facilitate reproducibility.
We modified appropriately the synthesis section. More details are also available in our previous work for equiatomic L10 CoPt nanoalloys (ref 44).
[44.] Tzitzios, V.; Niarchos, D.; Margariti, G.; Fidler, J.; Petridis, D. Synthesis of CoPt nanoparticles by a modified polyol method: characterization and magnetic properties. Nanotechnology 2005, 16, 287.
Please label each subfigure as a, b and c…
We thank the reviewer for the comment and the suggestion but we consider that it is not of so significant importance to replace (I), (II), (III) with (A), (B), (C). Additionally, all Figures in the manuscript are labelled as a, b and c except Figure 3. We avoid to use the A, B… labels in order to avoid any confusion with the lower case letters (a), (b) etc labels, which are also used in the same figure.
Reviewer 2 Report
Comments and Suggestions for Authors
The article contains interesting results on CoPt alloys. The article may published after the authors correct or explain the following points.
1) Lines 180-182. The authors introduce order parameter S and write an expression for it without any explanation. They need to explain why the order parameter is determined by the suggested expression.
2) Line 183-184. The authors write numerical value for anisotropy. How it was determined? The article does not contain any data to justify this conclusion.
Comments on the Quality of English LanguageEnglish may be improved.
Author Response
Reviewer 2:
The article contains interesting results on CoPt alloys. The article may published after the authors correct or explain the following points.
Lines 180-182. The authors introduce order parameter S and write an expression for it without any explanation. They need to explain why the order parameter is determined by the suggested expression.
We have used the expression:
Where is the ratio for the fully ordered phase and for the partially ordered phase [[i]]. The use of this expression is justified by the fact that the order parameter can be proved to scale where the parameter B can be considered independent of S [[ii]]
Line 183-184. The authors write numerical value for anisotropy. How it was determined? The article does not contain any data to justify this conclusion.
This is an approximate value considering that the anisotropy is roughly proportional to S [[iii],[iv]] and that the anisotropy of the fully ordered phase is 4.9MJ/m3.
[i] Y. S. Yu, T. A. George, W. L. Li, L. P. Yue, W. D. Fei, Haibo Li, Mei Liu, D. J. Sellmyer; “Effects of total thickness on (001) texture, surface morphology, and magnetic properties of [Fe/Pt]n multilayer films by monatomic layer deposition.” J. Appl. Phys. 1 October 2010; 108 (7): 073906. https://doi.org/10.1063/1.3489989
[ii] W Roberts, “X-ray measurement of order in CuAu, Acta Metallurgica, Volume 2, Issue 4, 1954,
Pages 597-603, https://doi.org/10.1016/0001-6160(54)90194-7.
[iii] Takayuki Kojima Masaki Mizuguchi, Tomoyuki Koganezawa, Keiichi Osaka, Masato Kotsugi, and Koki Takanashi 2012 Jpn. J. Appl. Phys. 51 010204 doi:10.1143/JJAP.51.010204
[iv] Yohei Kota, Akimasa Sakuma “Relationship between Magnetocrystalline Anisotropy and Orbital Magnetic Moment in L10-Type Ordered and Disordered Alloys”, Journal of the Physical Society of Japan, 81, 084705 (2012) 10.1143/JPSJ.81.084705
Reviewer 3 Report
Comments and Suggestions for Authors
The manuscript of Basina et al. is devoted to the microstructural, compositional and magnetic properties of Pt lean CoPt nanoparticles prepared polyol methodology. The objective of the work concerns the research of novel nanoparticles with high anisotropy requested as high dense recording media. L10 phase equimolar CoPt nanoparticles are one of the most promising materials. The study investigates the properties of lean Pt CoPt, around Co3Pt, considering the high costs and criticality of the Pt. The work explores the synthesis of these nanoparticles with polyol route followed by thermal annealing up to 700°C and several hours. The authors obtain interesting results: nanoparticles are multiphasic, composed by L10 CoPt and L2Co3Pt, with different exchange coupling mechanism what allows to obtain a coercive field of 1.4 T for nanoparticles annealed at 700°C for 4 hours.
The manuscript has a clear structure and complete characterizations. However, data and discussions are not well represented or supported. Magnetic measurements should be completed. Next questions should be considered:
TEM images: The quality of the manuscript could improve if High Resolution TEM images be included to illustrate the real structure and morphology of the particles.
Composition: Nominal composition of the nanoparticles is around Co3Pt1. In most of the text the presence of different phases with 1:1 and 3:1 Co:Pt are discussed. The presence of 1:1 phases implies the presence of Pt poor phases. Please, discuss these phases.
XRD: Explain the methodology of the analysis of the XRD spectra and the calculi of the structural data.
Represent the obtained structural data as a table.
In lines 195-199, the analysis of the XRD spectra indicates that the crystal size of the phases are similar to the TEM particle size. This conclusion is in contradiction with the fact that being each particle multiphase, grain size should be smaller than particle size. Clarify this point.
Magnetic properties:
The magnetic loops should be recorded to larger magnetic fields enough to saturate the samples. This will allow (a) to determine precisely the specific magnetization, (b) to determine the real coercive field and remanence of the samples and (c) to evidence experimentally the magnetic coupling or decoupling to exclude minor loops effects. Low temperature measurements could exclude that thermal demagnetization effects are not present at room temperature.
Explain more clearly the discussions. In particular:
- describe better the discussion on the magnetic properties of the as-prepared samples (lines 202-204). Other effects are possible like surface effects but also the presence of CoPt 1:1. This can increase the Hc.
- Lines 208 to 214. Why here?
-The discussion on the possible coupling mechanism must be quantitatively analyzed (page 7). The authors should consider a wider state of art on the coupling mechanisms in multiphase systems that consider not only spring magnet mechanism but also weak exchange or dipolar coupling.
Please, use IU units.
Author Response
Reviewer 3:
TEM images: The quality of the manuscript could improve if High Resolution TEM images be included to illustrate the real structure and morphology of the particles.
We understand the reviewer’s point but unfortunately, we are not able to have access and perform additional high resolution TEM measurements within a reasonable period of time.
Composition: Nominal composition of the nanoparticles is around Co3Pt1. In most of the text the presence of different phases with 1:1 and 3:1 Co:Pt are discussed. The presence of 1:1 phases implies the presence of Pt poor phases. Please, discuss these phases.
According to our estimations the nanoalloys composition is in the range of Co74.4Pt25.6. The nanoalloys possess both L10 and L12 crystal structures as evidenced by the XRD. The magnetic properties also denote the presence of a hard (L10) and a less hard phase (L12). However, is very difficult to estimate their percentage in the composite material.
In lines 195-199, the analysis of the XRD spectra indicates that the crystal size of the phases are similar to the TEM particle size. This conclusion is in contradiction with the fact that being each particle multiphase, grain size should be smaller than particle size. Clarify this point.
We agree with the reviewer, this is probably due to the shape irregularity of the multiphase materials after the extensive high temperature annealing.
Magnetic properties: The magnetic loops should be recorded to larger magnetic fields enough to saturate the samples. This will allow (a) to determine precisely the specific magnetization, (b) to determine the real coercive field and remanence of the samples and (c) to evidence experimentally the magnetic coupling or decoupling to exclude minor loops effects. Low temperature measurements could exclude that thermal demagnetization effects are not present at room temperature.
We fully agree with the reviewer’s point, however, our Vibrating Sample Magnetometer equipped with 2T and consequently we are not able to perform measurements with higher field in order to fully saturate the samples. This is the reason that we state clearly in the document that both magnetization and coercivity recorded at 2T field and are not the values that correspond to the saturation.
Explain more clearly the discussions. In particular:
- describe better the discussion on the magnetic properties of the as-prepared samples (lines 202-204). Other effects are possible like surface effects but also the presence of CoPt 1:1. This can increase the Hc.
We slightly modified the specific paragraph. We agree that the contribution of surface in these small sizes may be significant and cannot be excluded. However, in the case of these compounds that can exhibit various degrees of chemical ordering the tendency of Co atoms to move towards the surface can create compositional gradients and result in local chemical and associated magnetic anisotropy that does not match the structural long-range order parameter.
-The discussion on the possible coupling mechanism must be quantitatively analyzed (page 7). The authors should consider a wider state of art on the coupling mechanisms in multiphase systems that consider not only spring magnet mechanism but also weak exchange or dipolar coupling.
The microstructure revealed by the microscopy images is consistent with two types of interactions: Either exchange interactions within each particle for two-phase particles, or dipolar coupling between particles. The latter can be strong for neighbouring particles, but its sign depends on the relative orientation and therefore it is not always ferromagnetic. Thus, the effects of magnetostatic interactions can be quite complex and can assessed by the analysis of families of remanence curves (as in FORC or delta-M plots) using Preisach-type models which is out of the scope of the present work, simply focusing on the chemical method allowing to achieve high coercivity in low Pt-content Co-Pt based magnets.